# Improving Performance Prediction of Electrolyte Formulations with Transformer-based Molecular Representation Model

**Indra Priyadarsini** [1]   **Vidushi Sharma** [2]   **Seiji Takeda** [1]   **Akihiro Kishimoto** [1]   **Lisa Hamada** [1]   **Hajime Shinohara** [1]

## Abstract

Development of efficient and high-performing electrolytes is crucial for advancing energy storage technologies, particularly in batteries. Predicting the performance of battery electrolytes rely on complex interactions between the individual constituents. Consequently, a strategy that adeptly captures these relationships and forms a robust representation of the formulation is essential for integrating with machine learning models to predict properties accurately. In this paper, we introduce a novel approach leveraging a transformer-based molecular representation model to effectively and efficiently capture the representation of electrolyte formulations. The performance of the proposed approach is evaluated on two battery property prediction tasks and the results show superior performance compared to the state-of-the-art methods.

## 1. Introduction

Electrolytes are critical in many fields, including energy storage (batteries), fuel cells, sensors, and electrochemical devices. Despite their significance, designing efficient electrolytes and accurately predicting their performance is challenging due to the complex nature of electrolyte compositions and their interactions within battery systems (Sharma et al., 2023). The intersection of research in electrolyte formulation and machine learning (ML) represents an exciting frontier in materials science and technology development. Utilizing data-driven ML models allows for fast and accurate analysis of diverse factors related to electrolyte composition and structure, enabling the prediction and optimization of battery performance. Recent works on transformer-based Large Language Models (LLMs) for chemistry such

as (Wang et al., 2019; Chithrananda et al., 2020; Ross et al., 2022; Yüksel et al., 2023) have demonstrated significant strides in learning representations of chemical language from large unlabeled corpora and have shown great promise in molecular property prediction tasks. In most practical applications, individual molecules are a part of multi-constituent system, such as electrolyte formulations in batteries, that require capturing all individual constituents and their complex interactions to precisely predict the property or performance of the system. Thus, in this paper we introduce a transformer based approach suitable for multi-constituent systems such as battery electrolyte formulations.

## 2. Related Works

Constitute choice and their relative compositions make electrolyte formulation discovery a challenging multi-variate design problem. Integration of machine learning into electrolyte formulation research has the potential to revolutionize the field by speeding up the discovery and optimization of new complex materials. Recent works such as (Kim et al., 2023) adopt a data-driven approach using machine learning models to predict the Coulombic efficiency (CE) of lithium metal battery electrolytes by using elemental composition of the electrolytes as features to the ML model. However, the approach lacks generalizability across different formulations and formulation constituents. Further, (Sharma et al., 2023) introduced the Formulation Graph Convolution Network (F-GCN) model, designed to establish connections between the structure and composition of formulation components and the overall properties of liquid formulations. This model employs multiple graph convolution networks (GCNs) working in tandem to intuitively feature-engineer formulation components dynamically. Recently MM-Molformer, a transformer based approach was proposed (Soares et al., 2024). In MM-Molformer, the feature space was constructed by concatenating the embedding from Molformer model (Ross et al., 2022) along with the compositions of the constituents. Though the method showed significant improvement in the prediction results, the relation between the composition and the respective formulants cannot be guaranteed. Furthermore, both F-GCN and MM-Molformer required dummy featurization to maintain uniform feature size, thus making

---

[1]IBM Research - Tokyo, 19-21, Nihonbashi Hakozaki-cho, Tokyo 103-8510, Japan [2]IBM Almaden Research Center, 650 Harry Rd, San Jose, California 95120, United States . Correspondence to: Indra Priyadarsini <indra.ipd@ibm.com>.

*Accepted at the 1st Machine Learning for Life and Material Sciences Workshop at ICML 2024.* Copyright 2024 by the author(s).

the model not versatile to formulations having variable number of electrolyte components. In this paper we propose a novel approach to effectively capture the representation of electrolyte components, proportionate to their composition in the electrolyte formulation, to improve the performance of property prediction of electrolytes.

## 3. Method

Just like any other transformer-based models, the proposed approach also has two phases, pretraining and finetuning. In the pretraining phase, the model is trained on large unlabeled corpora in a self-supervised manner, with the goal of learning a robust molecular representation. In the finetuning phase, the molecular representation is used as input features for specific tasks such as property prediction. However since electrolyte formulation typically consists of two or more constituents, we introduce a feature construction phase to design a feature vector that captures the representation of the formulation. In this section, we describe the proposed approach in three parts - pretraining, feature construction, and finetuning.

### 3.1. Pre-training

The pre-training phase builds upon the transformer based large scale molecular representation model proposed in (Priyadarsini et al., 2023). A Bidirectional Auto-Regressive Transformer (BART) (Lewis et al., 2019) is pretrained on a mixture of 500M and 118M samples from the ZINC (Tingle et al., 2023) and PubChem (Kim et al., 2016) datasets, respectively. The dataset comprises of molecules represented in SMILES (Simplified Molecular Input Line Entry System) - a molecular string representation (Weininger, 1988). One of the drawbacks of SMILES is that it does not guarantee syntactic and semantic validity of the molecule (Krenn et al., 2022), especially when trained in autogenerative models such as BART, thus leading to a possiblity of learning invalid representations. To overcome this drawback, we pretrain our BART model with SELFIES (SELF-referencing

Embedded Strings). The SMILES strings are encoded to SELFIES representation as SELFIES provides a more concise and interpretable representation, making it suitable for machine learning applications where compactness and generalization are important (Krenn et al., 2022). The encoded SELFIES are tokenized using word level tokenization with a vocabulary size of 3160 tokens obtained from the ZINC and PubChem datasets. We then randomly mask 15% of the tokenized sequence and train the model autoregressively with a denoising objective. In comparison to encoder-only models, the BART model which is an encoder-decoder model has a better representation of the molecule owing to the denoising objective. Furthermore, since the model is trained on SELFIES instead of SMILES, the molecular representation i.e. encoder output is more robust and guarantees learning the representation of only valid molecules. Figure 1 shows the pretraining model architecture.

### 3.2. Feature construction

Figure 2. illustrates the schematic of the proposed method to construct a feature representation for a formulation. As seen from Figure 2.(a), the electrolyte formulation dataset consists electrolyte components and their corresponding concentration in the formulation. The concentrations are expressed as fractions of molar percentages. The formulations can be comprised of a mixture of any number of components. The first step in getting a good feature representation of this mixture is to get the molecular representations of the individual electrolyte components. The molecular representation is obtained from the pretrained BART model. Since the BART model is pretrained on SELFIES string, the electrolyte components are first encoded into SELFIES, tokenized and then fed to the pretrained BART model. The output of the encoder, i.e, molecular representation for each electrolyte component is a $d$ dimensional vector. Next, to capture the compositional information of each of the electrolyte components that make the formulation mixture, we scale the molecular representation of the electrolyte component with its corresponding concentration. The scaled vectors are then added to form an effective feature vector ($\mathbf{SA}$) of the formulation mixture. We hypothesize that this feature vector $\mathbf{SA}$ formed by the weighted linear combination of individual molecular representations that constitute the formulation can effectively capture overall representation of the electrolyte formulation. If $r_1, r_2, ..., r_n$ represent the molecular representation of the electrolyte components and $c_1, c_2, ..., c_n$ represent their corresponding composition, the resultant feature vector of the electrolyte formulation is given as,

$$\mathbf{SA} = c_1 r_1 + c_2 r_2 + ... + c_n r_n$$

where $n$ represents the number of electrolyte components, $c$ is a scalar and $r \in \mathbf{R}^d$. Since the scaled vectors are added,

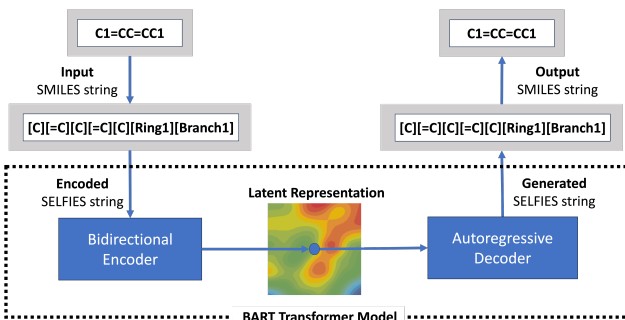

*Figure 1.* Pre-training model architecture

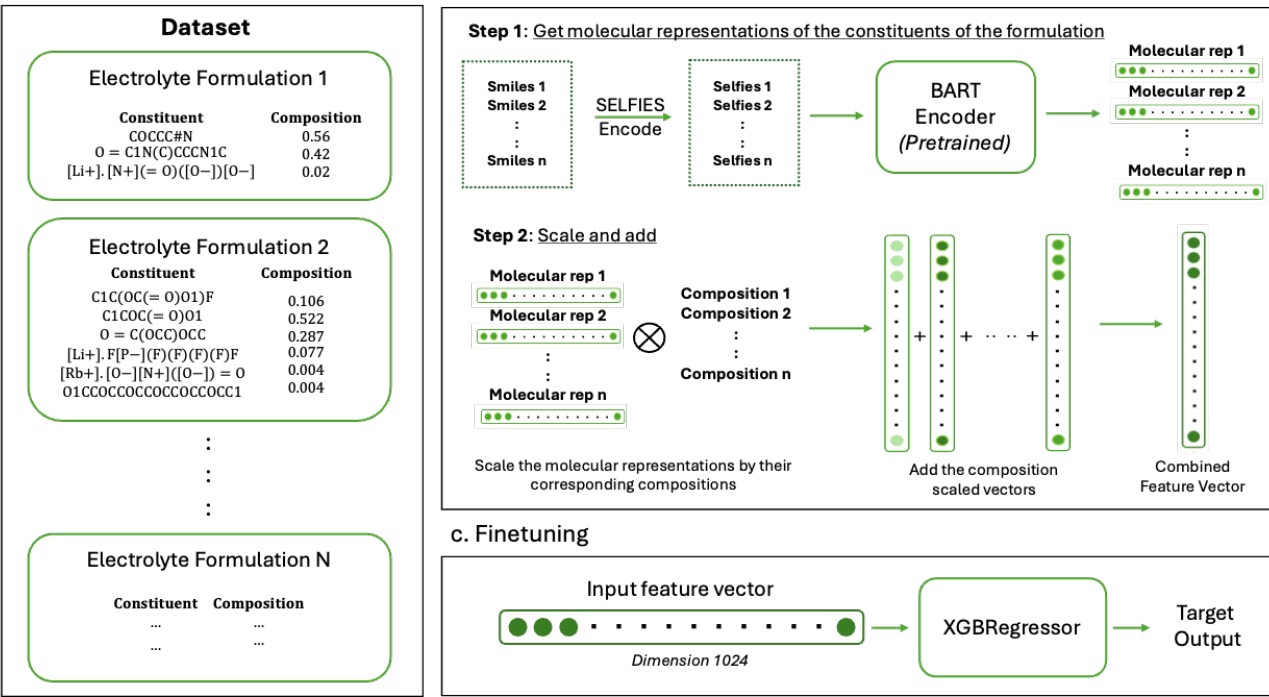

*Figure 2.* Illustration of the general schematic of the proposed method. (a) shows the general format of the electrolyte formulation dataset. (b) describes the procedure to construct the feature vector for an electrolyte formulation. (c) shows the fine-tuning model trained using the feature vector for a given prediction task.

the resulting feature vector is also a $d$ dimensional vector irrespective of the number of electrolyte components that constitute the formulation.

### 3.3. Fine-tuning

Finally, the feature vector (SA) obtained as a result of **S**caling **A**dding the molecular representations is used as the input feature vector in the finetuning or downstream models for tasks such as property prediction.

## 4. Results and Discussion

We evaluate the performance of the proposed approach on two datasets - Li—Cu half cell and Li—I full cell in the prediction of coulombic efficiency and specific capacities, respectively, given the electrolyte formulation. The datasets were randomly split in 80%-20% ratio for train and test samples. The input features are prepared as described in Section 3. We use the XGBoost algorithm (Chen & Guestrin, 2016) to train the prediction model. The metric chosen for performance evaluation is root mean squared error (RMSE), rounded off to 3 decimal places. We used Optuna (Akiba et al., 2019) for hyperparameter tuning and the results corresponding to the best hyperparameters are reported.

### 4.1. Coulombic Efficiency Prediction Task

Coulombic Efficiency defined as ratio of discharge and charge capacity, is a critical parameter in the study of battery performance and safety. The Li-Cu half cell dataset curated by (Kim et al., 2023) contains 147 entries of liquid electrolyte formulations along with their respective molar percentage and coulombic efficiency. Each electrolyte formulation entry comprises of 2 to 6 electrolyte components. Logarithmic Coulombic Efficiency (LCE) is used to numerically amplify the change in output with respect to the electrolytes. Table 1 shows the RMSE of the predicted LCE

| Method | RMSE |
|---|---|
| Linear regression (Kim et al., 2023) | 0.585 |
| Random forest (Kim et al., 2023) | 0.577 |
| Boosting (Kim et al., 2023) | 0.587 |
| Bagging (Kim et al., 2023) | 0.583 |
| F-GCN TL (Sharma et al., 2023) | 0.389 |
| MoLFormer (Soares et al., 2024) | 0.213 |
| MM-MoLFormer (Soares et al., 2024) | 0.195 |
| **BART-SA** | **0.148** |

*Table 1.* Comparison of RMSE of LCE prediction task

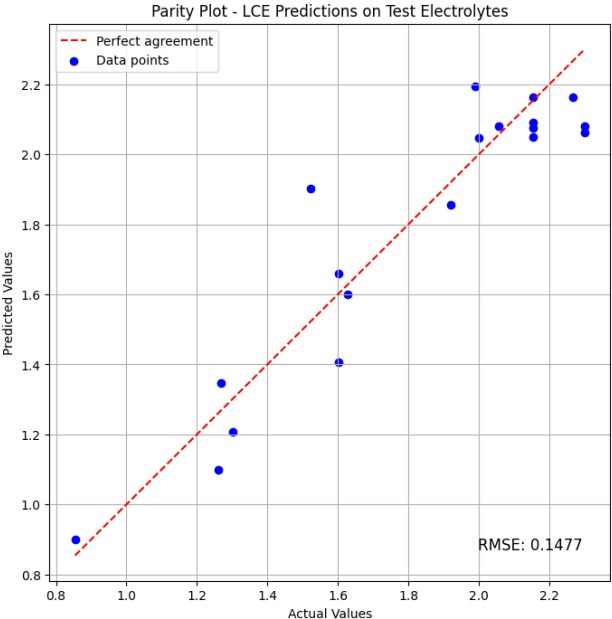

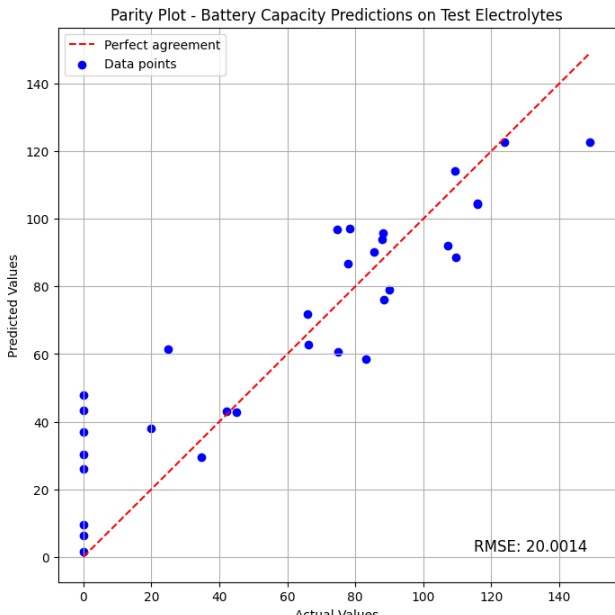

*Figure 3.* Parity plots showing predicted LCE values as scatterplots with respect to the actual values

*Figure 4.* Parity plots showing predicted battery capacities (in mAh/g) as scatterplots with respect to the actual values

values. The first four methods are based on elemental compositions proposed in (Kim et al., 2023), while F-GCN TL is a graph convolutional network with transfer learning. The MolFormer and MM-Molformer methods are transformer based methods pretrained on SMILES, and the feature vector is formed by concatenation of the molecular representation. Note that F-GCN TL, Molformer and MM-Molformer methods require dummy featurization to maintain a uniform feature size. As seen from the table, the proposed method BART-SA clearly outperforms existing methods with an RMSE value of 0.148. Figure 3 shows the parity plots of the LCE predicted and actual values. The predicted LCE values are in close approximation to the actual values.

### 4.2. Specific Capacity Prediction Task

Specific Capacity is a fundamental property of batteries that measures the amount of charge they can store per unit mass or volume. It is a critical parameter in evaluating the performance and suitability of different battery types for specific applications. The Li—I Full-Cell battery dataset was experimentally obtained by (Sharma et al., 2023) for Li-I battery coin cells with cycling tests at 1mA/cm$^2$. The dataset contains 125 entries of electrolyte formulations. Each electrolyte formulation comprises of 2-6 components. Table 2 shows the results of the proposed BART-SA method in comparison with the Formulation Graph Convolution Networks (F-GCN) with and without Transfer Learning (TL) from (Sharma et al., 2023). The BART-SA model had the lowest

specific capacity RMSE of 20.001 mAh/g. Figure 4 shows the parity plot of predicted and actual battery capacities in mAh/g. From the figure, it can be observed that while capacities > 30 mAh/g are well learnt while those with near zero specific capacities show more mispredictions. This is due to instability of battery cells with poor performing electrolyte formulations. Cells with capacity <30 mAh/g have high experimental uncertainties due to derogatory phenomenon like shuttling and parasitic reactions which continue to remain subject of investigation in LiI batteries (Giammona et al., 2023). The ability of the model to identify high performing electrolytes with good accuracies shows the potential of the current model in driving discovery of electrolyte formulations in high dimensional chemical space.

| Method | RMSE |
|---|---|
| F-GCN no-TL (Sharma et al., 2023) | 39.823 |
| F-GCN TL (Sharma et al., 2023) | 20.495 |
| **BART-SA** | **20.001** |

*Table 2.* Comparison of RMSE of Specific Capacity prediction task

## 5. Conclusion

This paper introduced a novel approach leveraging a transformer-based model to create a comprehensive feature vector for electrolyte formulations. This was achieved by obtaining molecular representations of individual electrolyte components from the pretrained BART model, scaling these

representations based on their respective concentrations in the formulation, and summing the scaled vectors to form a unified feature vector. This process ensured that the final feature vector effectively captured the compositional information of the formulation while maintaining a consistent dimensionality. The final feature vector was used in the evaluation of two battery property prediction tasks. The results showed superior performance compared to state-of-the-art methods. The superior performance can be speculated as a result of a better molecular representation obtained from the BART model pretrained with SELFIES. Further scaling the molecular representation with the concentration and adding aids to a better representation of the mixture as a whole. Overall, our proposed approach showed significant promise in advancing the field of electrolyte formulation by providing a robust, scalable, and efficient method for predicting properties.

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
