# OpenReview forum: "Improving Performance Prediction of Electrolyte Formulations with Transformer-based Molecular Representation Model"
_ICML.cc/2024/Workshop/ML4LMS — ML4LMS Poster_

### Official Review · Reviewer_Be6z · 2024-06-04
**Prediction of electrolyte formulations using data-driven molecular representation; Easy to follow; meets the standard of the workshop**

**Rating:** 7
**Confidence:** 5

**Review:**

This paper describes a data-driven approach to define a feature vector from smile strings (with several transformations and ML-learned representations) that are then used to predict several properties related to battery performance and safety.

I did not find this paper to be quite novel per se but given the goal of this workshop, I do not have any reservations about not accepting this paper.

---

### Official Review · Reviewer_tb5r · 2024-06-12
**Review for: Improving Performance Prediction of Electrolyte Formulations with Transformer-based Molecular Representation Model**

**Rating:** 6
**Confidence:** 4

**Review:**

-